# Contact tracing is an imperfect tool for controlling COVID-19 transmission and relies on population adherence

Emma L. Davis [1,6] ✉, Tim C. D. Lucas [1,6], Anna Borlase [1], Timothy M. Pollington [1,2], Sam Abbott[3], Diepreye Ayabina[1], Thomas Crellen [1], Joel Hellewell[3], Li Pi[1], CMMID COVID-19 Working Group*, Graham F. Medley [4], T. Déirdre Hollingsworth [1,7] & Petra Klepac [3,5,7]

Emerging evidence suggests that contact tracing has had limited success in the UK in reducing the R number across the COVID-19 pandemic. We investigate potential pitfalls and areas for improvement by extending an existing branching process contact tracing model, adding diagnostic testing and refining parameter estimates. Our results demonstrate that reporting and adherence are the most important predictors of programme impact but tracing coverage and speed plus diagnostic sensitivity also play an important role. We conclude that well-implemented contact tracing could bring small but potentially important benefits to controlling and preventing outbreaks, providing up to a 15% reduction in R. We reaffirm that contact tracing is not currently appropriate as the sole control measure.

[1] Big Data Institute, University of Oxford, Oxford, UK. [2] MathSys CDT, University of Warwick, Coventry, UK. [3] Department of Infectious Disease Epidemiology, London School of Hygiene and Tropical Medicine, London, UK. [4] Centre for Mathematical Modelling of Infectious Disease & Department of Global Health and Development, London School of Hygiene and Tropical Medicine, London, UK. [5] Department for Applied Mathematics and Theoretical Physics, University of Cambridge, Cambridge, UK. [6] These authors contributed equally: Emma L. Davis, Tim C. D. Lucas. [7] These authors jointly supervised this work: T. Déirdre Hollingsworth, Petra Klepac. *A list of authors and their affiliations appears at the end of the paper. ✉email: emma.davis@bdi.ox.ac.uk

In December 2019, SARS-CoV-2, a novel coronavirus strain, was detected in Hubei Province, China[1]. By 31 January 2020 the first UK cases of COVID-19, the disease caused by SARS-CoV-2, were confirmed[2]. Initial modelling studies indicated that fast and effective contact tracing could contain the UK outbreak in most settings[3,4]. However, by 20 March there were almost 4000 confirmed cases nationwide[5], at which point the UK Government scaled up physical distancing measures, including the closure of schools and social venues, extending to heightened restrictions on non-essential travel, outdoor activities and between-household social mixing[6]. Similar patterns occurred in other countries[7,8].

Throughout the pandemic, it has become obvious that the UK's NHS Test and Trace programme has not been as effective at reducing transmission as originally hoped, with a recent financial report even suggesting it has made no "measurable difference" to the course of the pandemic[9]. Results from the Department of Health and Social Care (DHSC) are more flattering, but still only conclude that contact tracing efforts reduced the R number by 2–5% in October 2020[10]. Speed of testing and tracing, poor integration with local authorities and adherence to isolation have all been cited as possible reasons. The head of NHS Test and Trace revealed in February 2021 that around 20,000 people a day had ignored isolation rules, despite being contacted by them[11]. This number is based on 80% adherence to isolation, which is optimistic compared to estimates from other surveys[12–14], and hence may be a substantial underestimate.

The UK's Test and Trace strategy has been continually updated throughout the pandemic, but with minimal, if any, observed improvement in results. A few elements have remained consistent throughout, such as traced contacts only being allowed to access tests once symptomatic and requiring a positive test result before tracing an isolated individual's contacts[15]. Current methods are reliant on PCR testing, but there is growing discussion around potential use of rapid lateral flow tests[16], which are considered to be substantially less sensitive[17].

Imperfect adherence (encompassing both completely non-adherent and partially-adherent individuals) to isolation and reporting and the innate difficulties in identifying contacts will pose challenges for contact tracing, particularly in crowded urban settings[18]. Therefore, evaluating both the limitations of contact tracing and how to maximise its effectiveness could be crucial in preventing an exponential rise in cases, which might see contact tracing capacity rapidly exceeded and stricter physical distancing measures required[19].

Extending Hellewell et al.'s[3] UK-focused contact tracing study with new insights could inform contact tracing strategy. Their key conclusion was that highly effective contact tracing would be sufficient to control an initial outbreak of COVID-19 in the UK, however subsequent evidence supports much higher pre- and asymptomatic transmission rates than had initially been considered. In particular, the original analysis only considered scenarios with 0–10% of cases as asymptomatic and 0–30% of transmission from symptomatic individuals occurring pre-symptoms, compared to more recent estimates of 31–42.5%[20–22] and 44%[23] respectively. The focus on speed in the UK contact tracing programme also requires a detailed assessment of the associated trade-offs through mechanistic modelling of the testing process. Up-to-date modelling studies are therefore needed to investigate the feasibility of contact tracing and the conditions under which it is effective.

We use improved incubation period and serial interval estimates[23–25], consider a range of self-reporting, adherence and tracing rates and simulate the use of diagnostic tests. We explore trade-offs between testing speed and sensitivity, and investigate the limitations of contact tracing. We conclude in which scenarios these methods are likely to be most effective.

## Results

The underlying assumptions around the contact tracing logistics that have been modelled and presented here are described in further detail in the Methods; see below for a detailed schematic of the contact tracing process.

**Efficacy of contact tracing.** We used data from a UK-based survey to consider three adherence scenarios[12]: a scenario with low self-reporting and poor adherence to isolation (representative of the small proportion of individuals who reported being fully adherent to advice from Test and Trace); a scenario with good reporting and adherence (representing those intending to be adherent to advice); and a scenario with good reporting and boosted adherence (where additional incentives to adhere to isolation are introduced).

Self-reporting and adherence can have a moderate-to-substantial impact on the efficacy of Test and Trace (Fig. 1). In the poor reporting & adherence scenario, there is no observable benefit of scaling up the coverage of contact tracing, or speeding up tracing. Increasing adherence and reporting to intended compliance levels shows a clear benefit to increasing contact tracing, with reductions in the outbreak risk and effective reproduction number R (dependent on the proportion of contacts traced and the magnitude of delays in testing and/or tracing) (Figs. 1 and 2).

However, even when Test and Trace is able to identify and trace 80% of contacts and compliance is good (good self-reporting, and adherence good or better), the associated reduction in R is only 6–13%. If coverage is lower, at around 40% of contacts, the reduction is 6% or below. This highlights the need for Test and Trace to be used as a supplemental measure, not as the sole control strategy.

**Diagnostic trade-offs.** Assuming good compliance to Test and Trace, we can investigate the comparative impact of a highly sensitive (95%) or poorly sensitive (65%) test (Fig. 3). There is a clear benefit to having a test with high sensitivity, and the relative benefit increases with contact tracing coverage.

Although there is an observable benefit to increasing the speed of testing and tracing, a two-day decrease in time to trace and obtain a test result, representing the use of a rapid test, appears insufficient to make up for switching from a 95% sensitive to a 65% sensitive test. However, the difference in probability of a large outbreak between instant testing with a 65% sensitive test and a two-day delay with a 95% sensitive test is relatively small. Hence conclusions around preferred test usage are likely to be sensitive to small changes in testing delay or sensitivity.

**Outbreak thresholds for contact tracing.** By considering the total number of observed and unobserved cases in an outbreak so far (Fig. 4) we can evaluate the probability of a large outbreak for a range of scenarios. It is hence possible to set thresholds for contact tracing feasibility. For example, consider a scenario with average reporting & adherence where 80% of contacts could be successfully traced. Then if, for example, the desired outcome was keeping the probability of a large outbreak below 50%, increasing testing and tracing speed from 2 days to instantaneous would raise the threshold number of cases from 166 to 357—an increase of 115%.

Improving contact tracing coverage has a visible effect in all scenarios presented, aside from those with poor reporting and adherence (left column). In scenarios with average or boosted adherence for a fixed number of total cases so far, speeding up tracing reduces the probability of a large outbreak and increases the relative benefit of higher contact tracing coverage.

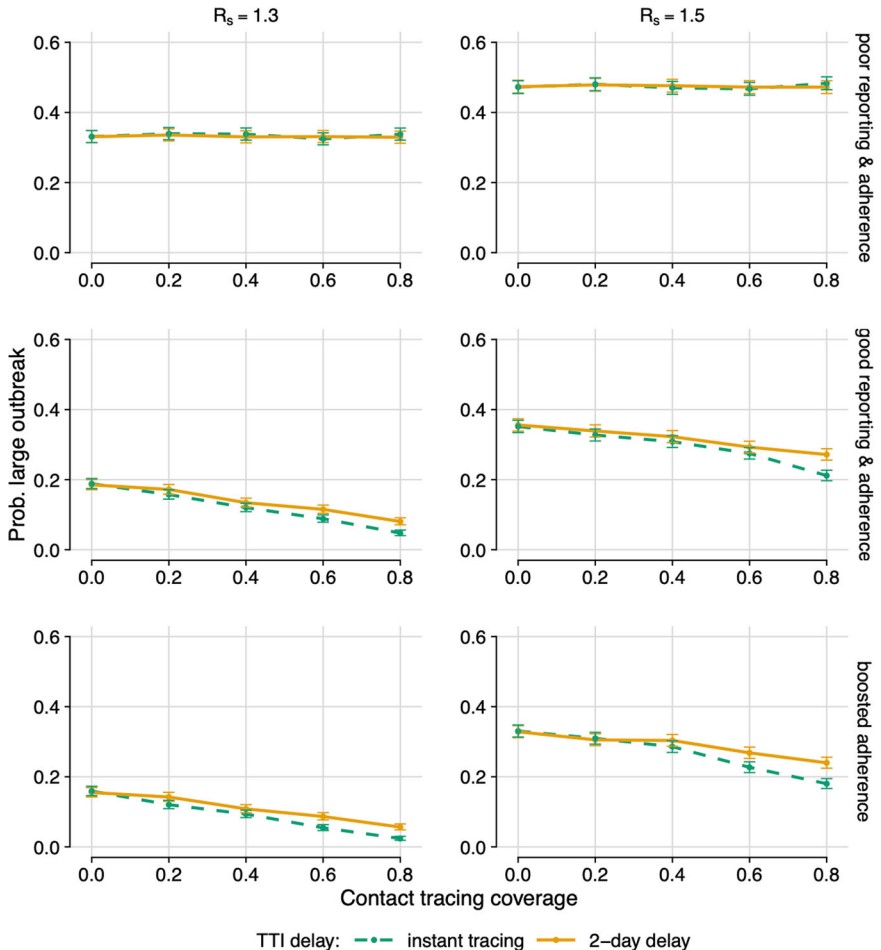

**Fig. 1 Probability of a large outbreak.** Probability of a large outbreak (>2000 cases) for different TTI (Test, Trace and Isolate) compliance scenarios for instant testing and tracing (dashed, green) and a 2-day delay (solid, orange), assuming 95% test sensitivity. Poor reporting & adherence (top): 11.9% self-reporting; 18.2% isolation on symptoms; 10.9% isolation on tracing. Good reporting & adherence (middle): 50% self-reporting; 70% isolation on symptoms; 65% isolation on tracing. Boosted adherence (bottom): 50% self-reporting; 70% isolation on symptoms; 90% isolation on tracing. Left: $R_s = 1.3$. Right: $R_s = 1.5$. Error bars: 99% confidence intervals from output variation of 5000 simulations. TTI test, trace and isolate.

## Discussion

Our results provide insights into why contact tracing implementation has not been as effective in reducing transmission as initially hoped in the UK. We conclude that the largest likely factor is adherence to the various stages of contact tracing and isolation, which is believed to be relatively poor[12]. However, if reasonable reporting and adherence can be achieved, then contact tracing efficacy could be improved by increasing the speed of test and tracing or increasing the proportion of contacts traced in the majority of scenarios considered. There are substantial behavioural challenges around improving adherence, requiring consideration of why individuals do not, or cannot, adhere to guidance. Isolation of traced contacts and positive-testing individuals is already legally required in England, so interventions would need to be targeted around messaging, logistical support and financial incentives. Even if good reporting and adherence, fast testing and high coverage of contact tracing can be achieved, our results demonstrate that the potential reduction in R is only around 10–15%. This confirms the emerging conclusion in the field that the UK contact tracing programme, in its current form, is ill-suited as the sole control strategy. However, more targeted investigative contact tracing could still be beneficial for identification and monitoring of new virus strains.

At this stage of the pandemic, there are more diagnostic tools available than in the initial months. In particular, use of rapid lateral flow device (LFD) tests is growing due to increased speed and reduced costs compared to PCR alternatives. However, our results suggest that test sensitivity is still important, with a 2-day 95% sensitive test performing better than an instantaneous 65% sensitive test in our model. This effect is seen under the assumption that negatively-testing individuals are not immediately released from isolation, but if they were then we would expect an even greater comparative loss of efficacy for the faster lower-sensitivity test. It is therefore vital to carefully consider the implications of changes in the testing methods employed in contact tracing.

Our model assumes constant test sensitivity across an individual's infected period, whereas a previous study shows that testing too early or late after exposure can dramatically increase false negative rates[26]. While assuming a fixed incubation period of 5 days, we have ignored temporal variation. Additionally, high between-person variance has been observed in the natural history of infection[23]. It is therefore unclear what drives these temporal changes in sensitivity or whether this temporal profile makes sense on an individual basis. These simplifying assumptions mean we may be over-estimating operational test sensitivity in some cases, leading to more optimistic results around the impact of contact tracing. This reinforces the conclusion that contact tracing is not currently appropriate as the sole control measure.

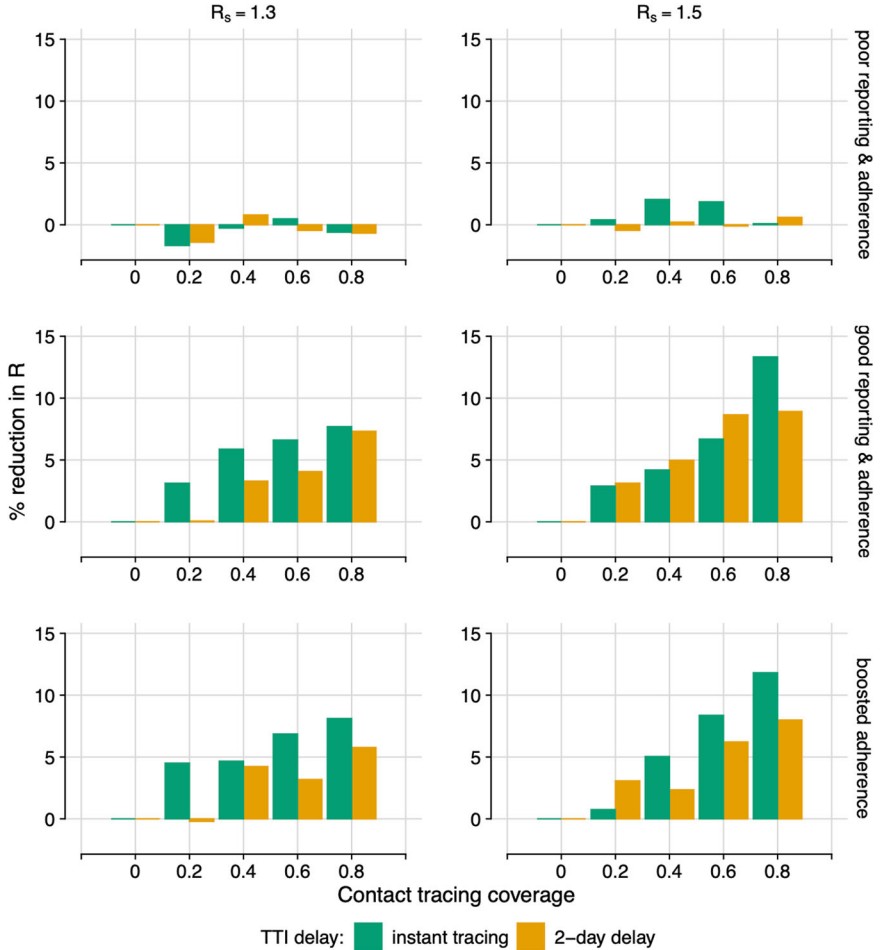

**Fig. 2 Contact tracing efficacy.** Percentage reduction in the effective reproductive number, *R*, for different Test and Trace compliance scenarios. For TTI (Test, Trace and Isolate) scenarios representing instant tracing (green) or a 2-day delay (orange). Assuming 95% test sensitivity. Poor reporting & adherence (top): 11.9% self-reporting; 18.2% isolation on symptoms; 10.9% isolation on tracing. Good reporting & adherence (middle): 50% self-reporting; 70% isolation on symptoms; 65% isolation on tracing. Good reporting & high adherence (bottom): 50% self-reporting; 70% isolation on symptoms; 90% isolation on tracing. Left: $R_s = 1.3$. Right: $R_s = 1.5$. Combined results of 5000 simulations. Negative values can occur due to stochastic fluctuation in the case of very low percentage change but do not represent contact tracing having a negative impact on transmission.

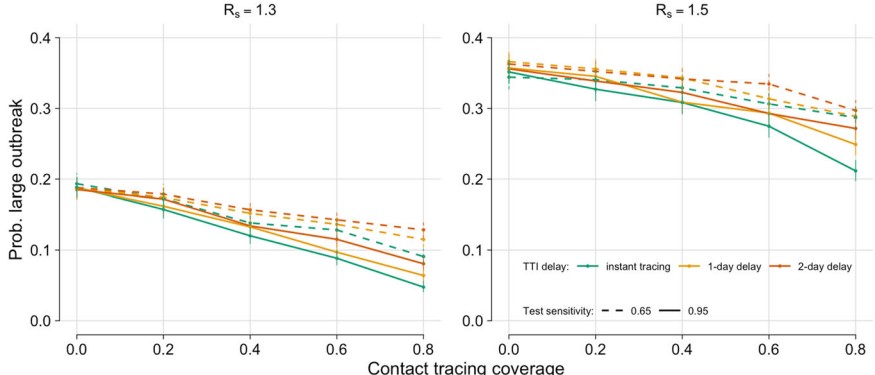

**Fig. 3 Diagnostic trade-offs.** Probability of a large outbreak (>2000 cases), by contact tracing coverage, for TTI (Test, Trace and Isolate) scenarios representing instant testing and tracing (green), a 1-day delay (orange) and a 2-day delay (red) with either 65% test sensitivity (dashed) or 95% (solid). Left: $R_s = 1.3$. Right: $R_s = 1.5$. Assuming good compliance (50% self-reporting, 70% isolation on symptoms, 65% isolation on tracing). Error bars: 95% confidence intervals from output variation of 5000 simulations.

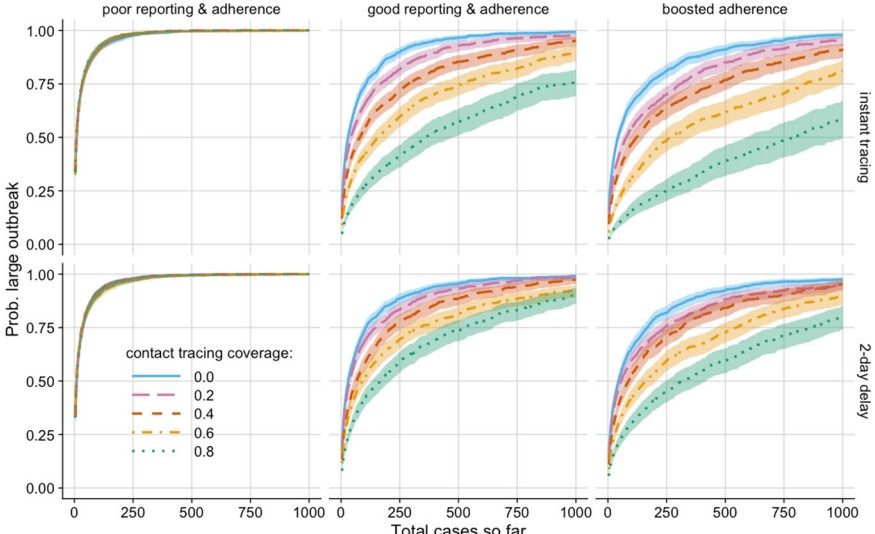

**Fig. 4 Outbreak thresholds.** Probability of a large outbreak (>2000 cases) by total number of cases so far (observed and unobserved). Sensitivity = 95%, self-reporting proportion = 50%, time to test from isolation = 1 day, with contact tracing coverage scenarios of 0 (blue, solid), 0.2 (pink, long-dashed), 0.4 (red, short-dashed), 0.6 (orange, dot-dashed) and 0.8 (green, dotted). Error windows: 95% confidence intervals from output variation of 5000 simulations.

We also assume the Negative Binomial dispersion, $k = 0.23$[27], of secondary cases, does not vary with $R_S$ due to different social distancing measures. This relationship is poorly characterised, but it is believed that social distancing may increase $k$, leading to decreased heterogeneity in number of contacts across the majority of the population due to an overall reduction in mean contacts, paradoxically making outbreak control harder, although this effect is expected to be cancelled out by the reduction in the mean[28]. Furthermore it is also possible that less heterogeneity in contacts may make tracing of individual contacts more feasible, allowing for a higher coverage.

The vaccine roll-out is currently in progress in the UK, with over 50% now believed to have COVID-19 antibodies through either vaccination or prior infection as of the end of March 2021[29]. This will have the effect of reducing $R$ and, eventually, when antibodies are sufficiently high, contact tracing may become viable as the sole control measure for keeping $R$ below 1.

Contact tracing improvements could include secondary contact tracing as seen in Vietnam, i.e. tracing the contacts of contacts of known cases, to get ahead of the chain of transmission[30]. Backwards contact tracing, whilst highly labour intensive, could also fill vital gaps where transmission links have been missed by focusing on tracing back from known cases to identify parent cases and potential super-spreaders[31]. As experience in contact tracing develops, it may be possible to give contacts a prior probability of infection (e.g. based on the contact duration and setting) and combine this with test results to improve existing isolation protocol. Testing of asymptomatic contacts would also allow tracing of currently hidden chains of infection, further reducing transmission.

Overall, we conclude that well-implemented contact tracing could bring small but potentially important benefits to controlling and preventing outbreaks, providing up to a 15% reduction in $R$. Reporting and adherence are the most important predictors of programme impact but tracing coverage and speed also play an important role, as well as diagnostic sensitivity. In line with a previous study[8], we have demonstrated that contact tracing alone is highly unlikely to prevent large outbreaks unless used in combination with evidence-based physical distancing measures, including restrictions on large gatherings.

## Methods

In this extension of a previous COVID-19 branching process model[3], the number of potential secondary cases generated by a primary case is drawn from a Negative Binomial distribution. The exposure time for each case, relative to infector onset, is drawn from a shifted Gamma distribution that allows for pre-symptomatic transmission and is left-truncated to ensure secondary case exposure time is after the primary case exposure time. Secondary cases are averted if the primary case is quarantined at the time of infection, assuming within household segregation is possible. The probability of quarantine depends on whether the primary case was traced, and adherence to self-isolation recommendations, irrespective of the test result (Fig. 5). Each simulation was seeded with five index cases that are initially undetected by the contact tracing system.

**Secondary case distribution.** A Negative Binomial distribution represented heterogeneity in onward transmission due to factors such as individual contact patterns or infectiousness, with the mean relating to the effective reproduction number under physical distancing $R_S$ (taking values 1.1, 1.3 or 1.5) with a constant dispersion parameter $k = 0.23$, taken from a study that used genome sequencing to investigate the clustering of secondary cases[27]. This also represents a mid-value among estimates, which vary widely from 0.1 (range: 0.05–0.2)[31] to 0.7 (range: 0.59–0.98)[32]. Here a smaller $k$ represents greater heterogeneity in transmission and results in the majority of index cases leading to no secondary infections, while a small proportion of individuals infect a large number of secondary cases. All parameter estimates and references can be found in Table 1.

**Generation interval.** The incubation period (time from exposure to symptoms) is assumed to follow a Lognormal distribution with mean 1.43 and standard deviation 0.66 on the log scale[24]. Each new case is then infected at an exposure time drawn from a Gamma-distributed infectivity profile (shape = 17.77, rate = 1.39 day$^{-1}$, shift = −13.0 days) relative to their infector's symptom onset. If this time is before the infector's exposure then this value is rejected and re-sampled to prevent negative generation intervals. This Gamma distribution has been fitted under these sampling assumptions to serial interval data published by He et al.[23] using the `fitdistr` package in R and our resulting distributions qualitatively match those presented in the original paper (see Supplementary Fig. S1 in Supplementary File for more details). The exposure time is then compared to the isolation times of the infector and cases are averted if the infector is in isolation when the infection event would have happened. For non-averted cases, symptom onset times are then drawn from the Lognormal incubation period distribution and the probability of a case remaining asymptomatic throughout their infected period is fixed at 31%[22].

**Contact tracing.** New cases are identified either through tracing contacts (like persons B & D in Fig. 5) of known cases or symptomatic individuals self-reporting to the system, which we model as a two-stage process. Firstly, if an individual is symptomatic (i.e. has a fever and/or dry persistent cough) but untraced we assume that a combination of reduced social activity due to illness, and awareness of COVID-19 prevention measures, results in a probability, $a_1$ of self-isolation one

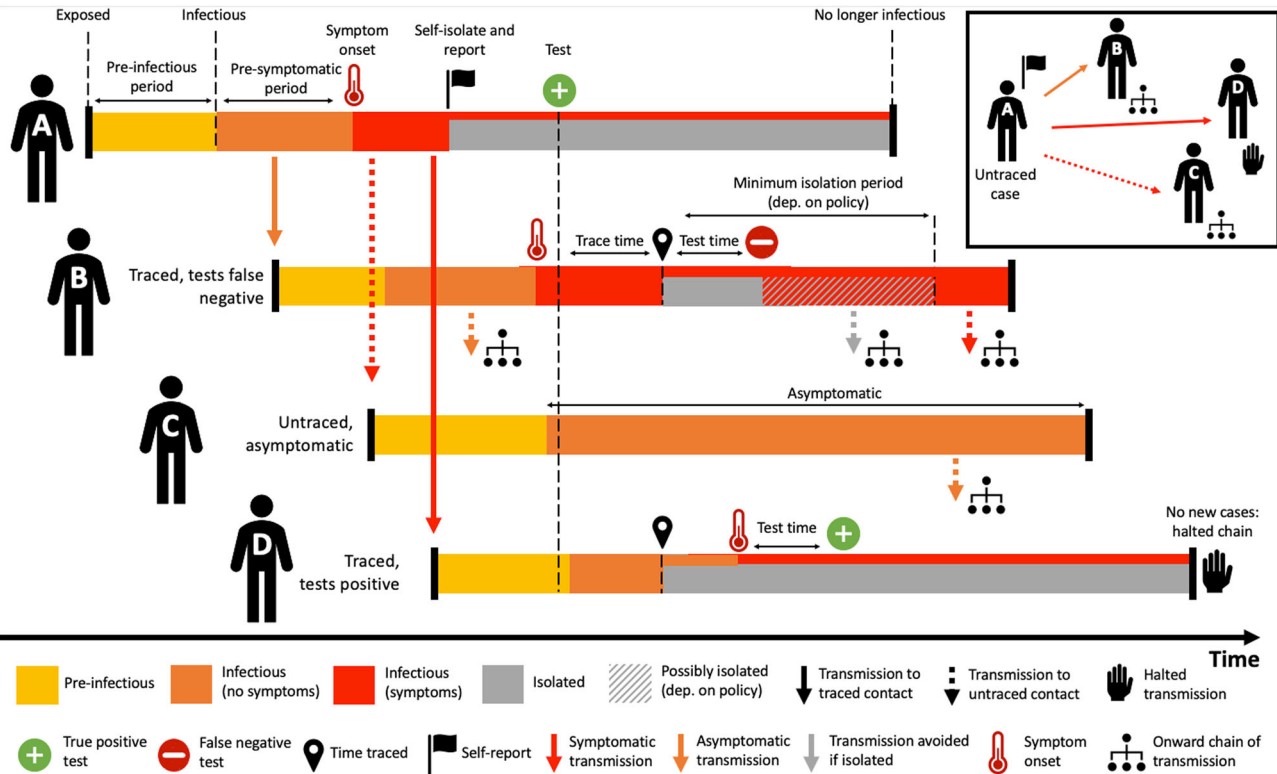

**Fig. 5 Contact tracing schematic.** Overview of the contact tracing process implemented in our model. Person A isolates and self-reports to the contact tracing programme with some delay after symptom onset, by which time they have infected Persons B, C and D. When Person A self-reports they isolate and are tested, a positive result initiates contact tracing. Person B was infected by A prior to their symptom onset and is detected by tracing after some delay. After isolating they are tested, with a false negative result. This leads to B either a) stopping isolation immediately or b) finishing a minimum 7 day isolation period. Both may allow new onward transmission. Person C was infected by A but not traced as a contact. Person C does not develop symptoms but is infectious, leading to missed transmission. Person D is traced and tested. The test for D returns positive, meaning that D remains isolated, halting this chain of transmission.

### Table 1 Model parameters values/ranges.

| Parameter | Values | Refs. |
|---|---|---|
| Number of initial cases | 5 | |
| Effective reproduction number under physical distancing, $R_S$ | 1.3, 1.5 | |
| Dispersion of $R_S$, $k$ | 0.23 | 27 |
| Proportion asymptomatic | 31% | 22 |
| Delay: onset to isolation | 1 day | |
| Incubation period (Lognormal) | Mean log: 1.43 | 23, 24 |
| Incubation period (Lognormal) | sd log: 0.66 | 23, 24 |
| Infection time (Gamma) | Shape: 17.77 | 23 |
| Infection time (Gamma) | Rate: 1.39 day$^{-1}$ | Fitted from[23] |
| Infection time shift | −13.0 days | 23 |
| Untraced self-isolation prob. | 18.2%, 70% | Based on[12] |
| Self-reporting probability | 11.9%, 50% | Based on[12] |
| Traced isolation adherence prob. | 10.9%, 65%, 90% | 12 |
| Contact tracing coverage | 0–80% | |
| Test sensitivity | 65%, 95% | 26, 35, 40, 41 |
| Time to test result (days) | 0, 2 | |
| Isolation duration if −ve test | 7 days | |

Parameters taken from the literature are fixed and for other parameters a range of values are explored.

day after symptom onset. Secondly, individuals who self-isolate in this way then have a probability, $a_2$, of contacting the tracing programme and reporting their symptoms as a potential case (like person A), which can be varied in the model.

Using data on UK adherence to the NHS Test and Trace programme from the CORSAIR study[12], we characterised three levels of population compliance. Firstly, as a lower bound, we considered the proportion of individuals who reported complete adherence to guidance: 18.2% reported adhering fully to isolation following onset of symptoms; 11.9% of symptomatic individuals self-reported their case to Test and Trace; and 10.9% of traced individuals isolated for the recommended duration. These reported figures were substantially lower that the intentional adherence reported by individuals who had not yet developed symptoms or been traced, which was taken as a good compliance scenario: 70% individuals said they would isolate following symptoms; 40–50% would self-report following onset; and 65% would isolate for the full duration if contacted by Test and Trace. We also considered a scenario with a boosted adherence to tracing of 90%.

Contact tracing is initiated when an existing case has been identified, isolated and returned a positive test (person A). The time taken to get a test result is either instant, 1 day or 2 days. The contacts of that individual are then traced with 40%–100% coverage. If a contact is successfully traced they will isolate with probability $a_3$. This continues until either the outbreak exceeds 2000 cases, or there are no further cases, resulting in disease extinction.

**Testing.** In simulations that include testing, we assume constant test sensitivities of 65% or 95%. The lower value aims to represent tests with poorer sensitivity, such as the rapid lateral flow tests that are seeing increased usage in 2021, based on estimates in the literature of 50.1–79.2% sensitivity[33,34]. The higher value represents the PCR tests that are available, with an estimated 94.8% sensitivity[34,35]. Due to the nature of the branching process model, only infected individuals are modelled so the impact of test specificity cannot be assessed under these methods, although the implications would be related to programme feasibility rather than efficacy. Current specificity estimates for both types of test are believed to be reasonably high in comparison[33,34,36–38], with some estimates of close to 100%, but false positive tests could lead to unnecessary negative socioeconomic impact under any scheme requiring quarantine of healthy individuals.

When testing is included in the model, all individuals that either self-report to the contact tracing system (person A in Fig. 5), or are traced contacts (persons B & D), are tested. From the moment a contact self-reports or is identified through tracing, either a zero-, one- or two-day delay is simulated before the test result is returned, chosen to be representative of UK programme targets. If a positive test is

returned, the individual's contacts are traced. If a negative test is returned, the individual is asked to complete a precautionary quarantine period of 7 days from the beginning of isolation.

**Simulation process**. Results presented are the combined output of 5000 simulations for each parameter combination or scenario, and each simulation is run for a maximum of 300 days. These results are used to derive the probability of a large outbreak given a range of conditions. A large outbreak is defined as 2000 cases: this threshold was chosen from experimental runs with a maximum of 5000 cases and noting which of the simulated epidemics went extinct; 99% of extinctions occurred before reaching 2000 cases. The model was written in R with pair code review and unit tests[39]. The code is available from a public GitHub repository (www.github.com/timcdlucas/ringbp).

**Reporting summary**. Further information on research design is available in the Nature Research Reporting Summary linked to this article.

## Data availability
The data that support the findings of this study are available in Zenodo via https://doi.org/10.5281/zenodo.4752369.

## Code availability
The code used in this study is available in Zenodo via https://doi.org/10.5281/zenodo.4752369.

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

## Acknowledgements
E.L.D., T.C.D.L., A.B., D.A., L.P., T.M.P., G.F.M. & T.D.H. gratefully acknowledge funding of the NTD Modelling Consortium (NTDMC) by the Bill & Melinda Gates Foundation (BMGF) (grant no. OPP1184344). E.L.D., L.P. & T.D.H. gratefully acknowledge funding from the MRC COVID-19 UKRI/DHSC Rapid Response grant MR/V028618/1 and JUNIPER Consortium (MR/V038613/1). The following funding sources are acknowledged as providing funding for the named authors. This research was partly funded by the Bill & Melinda Gates Foundation (NTDMC: OPP1184344: G.F.M.). This project has received funding from the European Union's Horizon 2020 research and innovation programme - project EpiPose (101003688: P.K.). Royal Society (RP/EA/180004: P.K.). Wellcome Trust (210758/Z/18/Z: J.H., S.A.).

Views, opinions, assumptions or any other information set out in this article should not be attributed to BMGF or any person connected with them. T.C. is funded by a Sir Henry Wellcome Fellowship from the Wellcome Trust (215919/Z/19/Z). T.M.P.'s PhD is supported by the Engineering & Physical Sciences Research Council, Medical Research Council and University of Warwick (EP/L015374/1) and thanks Big Data Institute for hosting him. All funders had no role in the study design, collection, analysis, interpretation of data, writing of the report, or decision to submit the manuscript for publication.

## Author contributions

Conceptualisation: E.L.D., T.C.D.L., P.K., G.F.M., T.D.H. Formal Analysis: E.L.D., T.C.D.L. Funding acquisition: T.D.H. Investigation: E.L.D., T.C.D.L., A.B., T.M.P., L.P., D.A., T.C. Methodology: E.L.D., T.C.D.L. Software: E.L.D., T.C.D.L., S.A., J.H. Validation: E.L.D., T.C.D.L. Visualisation: E.L.D., A.B. Writing - original draft: E.L.D., T.C.D.L. Writing - review & editing: All authors.

## Competing interests

The authors declare no competing interests.

## Additional information

## CMMID COVID-19 Working Group

Rachel Lowe[5], Akira Endo[5], Nicholas Davies[5], Georgia R. Gore-Langton[5], Timothy W. Russell[5], Nikos I. Bosse[5], Matthew Quaife[5], Adam J. Kucharski[5], Emily S. Nightingale[5], Carl A. B. Pearson[5], Hamish Gibbs[5], Kathleen O'Reilly[5], Thibaut Jombart[5], Eleanor M. Rees[5], Arminder K. Deol[5], Stéphane Hué[5], Megan Auzenbergs[5], Rein M. G. J. Houben[5], Sebastian Funk[5], Yang Li[5], Fiona Sun[5], Kiesha Prem[5], Billy J. Quilty[5], Julian Villabona-Arenas[5], Rosanna C. Barnard[5], David Hodgson[5], Anna Foss[5], Christopher I. Jarvis[5], Sophie R. Meakin[5], Rosalind M. Eggo[5], Kaja Abbas[5], Kevin van Zandvoort[5], Jon C. Emery[5], Damien C. Tully[5], Frank G. Sandmann[5], W. John Edmunds[5], Amy Gimma[5], Gwen Knight[5], James D. Munday[5], Charlie Diamond[5], Mark Jit[5], Quentin Leclerc[5], Alicia Rosello[5], Yung-Wai Desmond Chan[5], David Simons[5], Sam Clifford[5], Stefan Flasche[5], Simon R. Procter[5] & Katherine E. Atkins[5]

A full list of members and their affiliations appears in the Supplementary Information.

