## [Peer Review File · Nature Communications]

REVIEWER COMMENTS

Reviewer #1 (Remarks to the Author):

In this manuscript Davis et al. explore the complications that imperfect COVID-19 test sensitivity has on the impact of contact tracing. In particular, their continuous time branching process model is used to evaluate how a shorter quarantine time for false-negative cases can lead to increased transmission. They note that a faster turn-around time for testing ironically exacerbates this effect since false-negative cases are released from quarantine earlier in their infectious period.

Given the emphasis on testing as a means of controlling the COVID-19 pandemic, the manuscript's exploration of limitations of testing in the context of contact tracing is of obvious importance. In addition, I think the manuscript provides thorough analysis, based on sound methodological approaches. My biggest concern is that the manuscript reads as a number of crisscrossed storylines, rather than one cohesive story. Thus, I find it difficult to gauge its impact. As currently written, I think a typical reader might feel the manuscript simply supports well-known principles (e.g. outbreaks are more likely when R is high, and poor test sensitivity makes it impossible to adequately quarantine infectious individuals via a 'test and contain' strategy). However, I think that with a more focused and organized storyline the manuscript has the potential to communicate more novel insights, such as specific benchmarks for contact tracing programs to strive for in order to achieve successful control of transmission. Similarly, I think the public health messaging could be improved (e.g. without understanding the context, the abstract suggests that 'faster testing' is detrimental). Some more specific thoughts are listed below:

- I think the title can be improved as it is hard to make sense of what the paper is about from the title alone
- I would avoid such prominent emphasis on the specificity being 65% (particularly in the abstract and introduction). The meta-analysis [15] used to obtain this estimate appears to only use results from nasal and oral samples, rather than the more sensitive nasopharyngeal approach. Many would argue that the sensitivity is higher for properly acquired samples (e.g. ref [41]), and thus I would present results in a way that doesn't focus on the assumption of 65%. In addition, the effective sensitivity may be higher since symptomatic individuals who test negative will typically still be asked to quarantine under the presumption that they are infected.
- I think the abstract can be significantly improved to provide a clearer explanation of the focus of this paper and its implications for public health. As currently written, some may think the manuscript concludes that fast turn-around for tests is entirely disadvantageous.
- Line 49: Vague - Imperfect adherence to what?
- Line 61: A bit vague. I would provide additional detail about what level of pre-/asymptomatic transmission was previously assumed vs current estimates
- Line 73: I think figure 4 is really helpful for understanding the results and so I would start the results section with some reference to that figure (e.g. perhaps start by describing the various possible scenarios for which asymptomatic transmission can occur, despite contact tracing).

- Lines 82 and 374: Define testing delay. It's unclear if it refers to the delay until testing is obtained, or to the delay from the time a sample is obtained to the time the results are known.
- Line 94/401: I am not sold on 2,000 being used as a meaningful threshold for a large outbreak. I appreciate the explanation provided, but it strikes me as too heavily dependent on theory rather than applicability. Much before reaching a size of 2,000, it seems there would be changes to human behavior and decreased capacity for contact tracing. This would then upend the assumptions of constant parameters for the model. Perhaps a smaller threshold for a 'large outbreak' is warranted?
- Line 124: For figure 2, I found the 2x2 layout challenging for comparing the results for different values of the sensitivity.
- Line 142: For figure 3, I find it hard to understand the key message. Although it's titled 'superspreading scenarios', it seems that the main message of 3c/3d is that the probability of an outbreak increases as R increases, and contact tracing decreases.
- Lines 152/157: Regarding 'current outbreak size ... could be used to assess at what point during an epidemic contact tracing would be unable to control transmission, as well as to inform targets for coverage and speed' and 'We also found that higher contact tracing coverage results in a lower overall number of individuals which are traced, tested and quarantined.' I think these are interesting and to improve overall impact, I would consider making associated plots. Otherwise, I'd move these conceptual thoughts into the discussion.
- Line 254: It is unclear to me why social distancing would decrease heterogeneity. I'd actually expect the opposite if some groups of individuals social distance more than other groups.
- Line 265: I would define backward contact tracing.
- Line 273: I find that this summary statement is a better representation of the manuscript's findings than the current title and abstract. In general, I would think carefully about the overall message you want to provide readers focused on public health. In particular – do you want to focus on the potential benefit of contact tracing when used in parallel with other measures, or on the reasons why contact tracing might not work?
- Line 311: Somewhere in this section or in the figure 5 legend, I think it would help to explain the distinction between generation and serial interval.

Reviewer #2 (Remarks to the Author):

The authors used a previously published simulation model to assess the impact of test sensitivity on the effectiveness of contact tracing and quarantine. The main conclusion is that poor test sensitivity, which might lead to false negative tests and persons testing negative while they are still infectious, but leave quarantine might have an adverse impact on contact tracing. It all depends on the exact timing of various contact tracing steps.

Overall, a useful paper. My main concern is that I get the impression that test sensitivity was modelled as a constant instead of depending on time since exposure. I did not fully understand how test sensitivity was defined in the model. Both options are shown in figure 1a. If it is taken as constant, does this not overestimate the effectiveness of contact tracing?

The delays assumed in the simulations seem optimistic to me. Maybe good to show also less optimistic scenarios?

The definition of a large outbreak as more than 2000 cases seems high to me. I would expect that once the outbreak reaches a size of maybe 100, there would be hardly any extinction events. If this is not the case, can you give some more explanation? Is this due to the skewness of the contact distribution?

Page 4: In the superspreading events, do you take into account the delays between developing symptoms and getting hospitalized?

Section Additional observations: I don't really understand this idea of using the current outbreak size to say whether contact tracing can still control transmission. Do you mean here that the outbreak becomes too large for the public health system to perform the tracing? Theoretically, the tracing effectiveness does not depend on the initial number of individuals, but only on what proportion of onward infections can be prevented. Please clarify this.

Page 11, lines 380/381: I don't understand this sentence. Why should contacts of a negative testing person be isolated?

Table 1 and Figure 5: The parameters you use for infection time are different than the ones reported by He et al (your ref 23) originally. Are these the values for the corrected analysis? However, they are also different from the parameters reported by Ashcroft in the Swiss Medical Weekly (doi:10.4414/smw.2020.20336). Could you please comment on this and justify your use of parameter values?

Figures 1 and 2: Please leave more space between upper and lower panels or add a horizontal line at zero for the upper panel.

Figure 1a: From this figure I get the impression that you used a constant test sensitivity, not dependent on the time since infection. Is that correct? If not, I suggest to plot the test sensitivity you used in comparison to the Kurcika values. It is not clear to me from this figure, how this shift in testing was applied in relation to test sensitivity in the model. Please clarify.

Figure 1b: In the left panel, what does it mean for the line "no testing" that individuals leave quarantine after negative testing? Would individuals remain in quarantine when there is no testing? So is this exactly the same line as in the right panel for no testing? It looks to be somewhat lower. Can

you clarify this in the figure caption?

Figure 2: It might help the reader to include some gridlines or to add numbers to the medians of the boxplots. Now it is a bit hard to see how much higher or lower the plots are.

Figure 3: I am not sure whether first hospitalization would be the trigger to start contact tracing or other measures. Why not detection of first symptomatic case? I would mention in the figure caption that a large outbreak is defined as 2000 cases.

Figure 4: Nice overview over contact tracing process!

Reviewer #1 (Remarks to the Author):

In this manuscript Davis et al. explore the complications that imperfect COVID-19 test sensitivity has on the impact of contact tracing. In particular, their continuous time branching process model is used to evaluate how a shorter quarantine time for false-negative cases can lead to increased transmission. They note that a faster turn-around time for testing ironically exacerbates this effect since false-negative cases are released from quarantine earlier in their infectious period.

Given the emphasis on testing as a means of controlling the COVID-19 pandemic, the manuscript's exploration of limitations of testing in the context of contact tracing is of obvious importance. In addition, I think the manuscript provides thorough analysis, based on sound methodological approaches. My biggest concern is that the manuscript reads as a number of crisscrossed storylines, rather than one cohesive story. Thus, I find it difficult to gauge its impact. As currently written, I think a typical reader might feel the manuscript simply supports well-known principles (e.g. outbreaks are more likely when R is high, and poor test sensitivity makes it impossible to adequately quarantine infectious individuals via a 'test and contain' strategy). However, I think that with a more focused and organized storyline the manuscript has the potential to communicate more novel insights, such as specific benchmarks for contact tracing programs to strive for in order to achieve successful control of transmission. Similarly, I think the public health messaging could be improved (e.g. without understanding the context, the abstract suggests that 'faster testing' is detrimental). Some more specific thoughts are listed below:

We thank the reviewer for their positive assessment of the methodological approaches and analysis and agree that the message in the original manuscript did perhaps not present the most cohesive story. We have reformulated the message, partially guided by the changing global and UK situations, but also to ensure that the findings presented follow a more organized storyline. We have built on the idea of specific benchmarks for contact tracing, without being overly prescriptive, by considering the impact of variation in several key parameters, such as adherence to isolation and tracing/testing speed. We have also carefully considered the phrasing around the statement of results to ensure the public health context will not be misunderstood.

- I think the title can be improved as it is hard to make sense of what the paper is about from the title alone

We have reformulated the title (previously: "*An imperfect tool: COVID-19 'test & trace' success relies on minimising the impact of false negatives and continuation of physical distancing*"), to provide a clearer message on the impact of contact tracing and the conclusions around important factors.

New title: "*An imperfect tool: contact tracing could provide valuable reductions in COVID-19 transmission if good adherence can be achieved and maintained.*"

- I would avoid such prominent emphasis on the sensitivity being 65% (particularly in the abstract and introduction). The meta-analysis [15] used to obtain this estimate appears to only use results from nasal and oral samples, rather than the more sensitive naso-pharyngeal approach. Many would argue that the sensitivity is higher for properly acquired samples (e.g. ref [41]), and thus I would present results in a way that doesn't focus on the assumption of 65%. In addition, the effective sensitivity may be higher since symptomatic individuals who test negative will typically still be asked to quarantine under the presumption that they are infected.

We thank the reviewers for pointing out the emphasis on a 65% sensitivity test as taking a narrow view of the situation. We have reformulated the results to represent more up-to-date estimates of test sensitivity and now consider a lower sensitivity test (65%) and a higher sensitivity test (95%). These two levels of sensitivity have been chosen to match estimates of LFT and PCR test sensitivities reported in the literature (references 33, 34 and 35 in the revised manuscript).

- I think the abstract can be significantly improved to provide a clearer explanation of the focus of this paper and its implications for public health. As currently written, some may think the manuscript concludes that fast turn-around for tests is entirely disadvantageous.

We have rewritten the abstract to represent the new findings, with a focus on the importance of adherence to contact tracing measures. We have clearly summarised our new results and referenced their implications for contact tracing as a public health tool for controlling COVID-19 transmission.

- Line 49: Vague - Imperfect adherence to what?

Sentence has been updated to start: “Imperfect adherence to isolation and reporting...” to clarify the statement (now line 45).

- Line 61: A bit vague. I would provide additional detail about what level of pre-/asymptomatic transmission was previously assumed vs current estimates

This is a relevant and useful point, we have updated this section of the introduction to include estimates (lines 56-59).

- Line 73: I think figure 4 is really helpful for understanding the results and so I would start the results section with some reference to that figure (e.g. perhaps start by describing the various possible scenarios for which asymptomatic transmission can occur, despite contact tracing).

This is a good suggestion, we have referenced the figure (now Figure 5) at the beginning of the results.

- Lines 82 and 374: Define testing delay. It’s unclear if it refers to the delay until testing is obtained, or to the delay from the time a sample is obtained to the time the results are known.

We have updated discussions around time-to-test to make it clear that we are referring to the time taken from the identification of a contact to tracing and obtaining a test result for that contact (lines 101 and 289).

- Line 94/401: I am not sold on 2,000 being used as a meaningful threshold for a large outbreak. I appreciate the explanation provided, but it strikes me as too heavily dependent on theory rather than applicability. Much before reaching a size of 2,000, it seems there would be changes to human behavior and decreased capacity for contact tracing. This would then upend the assumptions of constant parameters for the model. Perhaps a smaller threshold for a ‘large outbreak’ is warranted?

We appreciate this comment around the potential changes in human behaviour caused by a growing outbreak, but have chosen to keep the 2,000 case threshold as it represents a sensible threshold beyond which stochastic extinction of the outbreak is highly unlikely. With the large scale-ups over the previous 6-12 months in testing and contact tracing capacity, we believe capacity is unlikely to be a substantial factor in model parameterisation at this case level. Although it is possible that some population behaviour change may occur, most behaviour change is currently driven by changes in regulations/restrictions rather than in numbers of cases. As we are considering the impact of contact tracing only, and holding other interventions – such as changes to social restrictions – as fixed, we believe our fixed parameter assumptions are still relevant for showing how changes in contact tracing might impact outbreak risk.

- Line 124: For figure 2, I found the 2x2 layout challenging for comparing the results for different values of the sensitivity.

We have kept the grid-layout of plots to demonstrate a range of parameter values but have added axes and gridlines to the individual plots for all figures to allow better comparison between different values/scenarios. We have demonstrated multiple scenarios on one plot where possible without compromising readability.

- Line 142: For figure 3, I find it hard to understand the key message. Although it’s titled ‘superspreading scenarios’, it seems that the main message of 3c/3d is that the probability of an outbreak increases as R increases, and contact tracing decreases.

We have removed 3a/b from this figure and instead expanded on 3c/d to demonstrate the changing probability of an outbreak for varying contact tracing coverage across different reporting/adherence and TTI speed scenarios (now Figure 4).

- Lines 152/157: Regarding ‘current outbreak size ... could be used to assess at what point during an epidemic contact tracing would be unable to control transmission, as well as to inform targets for coverage and speed’ and ‘We also found that higher contact tracing coverage results in a lower overall number of individuals which are traced, tested and quarantined.’ I think these are interesting and to

improve overall impact, I would consider making associated plots. Otherwise, I'd move these conceptual thoughts into the discussion.

We have removed this section to focus the results on the key messages around determinants of contact tracing success.

- Line 254: It is unclear to me why social distancing would decrease heterogeneity. I'd actually expect the opposite if some groups of individuals social distance more than other groups.

We have expanded on this point to explain that reducing the mean number of contacts for most individuals is likely to also lead to a reduction in heterogeneity across the majority of the population, although we agree it is hard to characterise (now lines 168-172)

- Line 265: I would define backward contact tracing.

We have expanded on this sentence to better define what is meant by backward contact tracing (now lines 182-185).

- Line 273: I find that this summary statement is a better representation of the manuscript's findings than the current title and abstract. In general, I would think carefully about the overall message you want to provide readers focused on public health. In particular – do you want to focus on the potential benefit of contact tracing when used in parallel with other measures, or on the reasons why contact tracing might not work?

We have taken the Reviewer's suggestions into account and reformulated the message to focus on the factors that impact contact tracing success, concluding that adherence to isolation and reporting is the most important determinant of the scope of contact tracing as a control measure.

- Line 311: Somewhere in this section or in the figure 5 legend, I think it would help to explain the distinction between generation and serial interval.

This figure has been moved into the Supplementary Information, where we have described the definitions of the generation and serial intervals for reference.

Reviewer #2 (Remarks to the Author):

The authors used a previously published simulation model to assess the impact of test sensitivity on the effectiveness of contact tracing and quarantine. The main conclusion is that poor test sensitivity, which might lead to false negative tests and persons testing negative while they are still infectious, but leave quarantine might have an adverse impact on contact tracing. It all depends on the exact timing of various contact tracing steps.

Overall, a useful paper. My main concern is that I get the impression that test sensitivity was modelled as a constant instead of depending on time since exposure. I did not fully understand how test sensitivity was defined in the model. Both options are shown in figure 1a. If it is taken as constant, does this not overestimate the effectiveness of contact tracing?

We thank the reviewer for their acknowledgement of the usefulness of this paper, and have taken measures to address the perceived lack of clarity around how testing was modelled. We have now specified clearly in the Methods section that we assume a constant test sensitivity, considering two scenarios with either 65% or 95% sensitivity.

We considered expanding the model to consider test sensitivity depending on time since exposure, but ultimately concluded that, although this is an important factor, it was beyond the scope of this manuscript. This does mean that our model may overestimate the effectiveness of contact tracing, which we have highlighted in the Discussion as follows:

“Our model assumes constant test sensitivity across an individual's infected period, whereas a previous study shows that testing too early or late after exposure can dramatically increase false negative rates [26]. While assuming a fixed incubation period of 5 days, we have ignored temporal variation. Additionally, high between-

person variance has been observed in the natural history of infection [24]. It is therefore unclear what drives these temporal changes in sensitivity or whether this temporal profile makes sense on an individual basis. These simplifying assumptions mean we may be over-estimating operational test sensitivity in some cases, leading to more optimistic results around the impact of contact tracing. This reinforces the conclusion that contact tracing is not currently appropriate as the sole control measure.”

The delays assumed in the simulations seem optimistic to me. Maybe good to show also less optimistic scenarios?

We have considered two lengths of delay: an instant test, representing LFTs, and a test with a 2-day delay, representing the slower PCR test, to allow comparison between different testing strategies. We believe 48 hours is a reasonable turnaround time assumption for the PCR test. We did consider scenarios with longer delays, but we saw no qualitative patterns that cannot be observed by comparing 0- and 2-day delays and therefore decided including further scenarios didn't add to the message of the manuscript.

The definition of a large outbreak as more than 2000 cases seems high to me. I would expect that once the outbreak reaches a size of maybe 100, there would be hardly any extinction events. If this is not the case, can you give some more explanation? Is this due to the skewness of the contact distribution?

Yes, this is due to the heterogeneity / dispersion in the secondary case distribution, as most primary cases lead to no new infections. Although an outbreak size of around 250 does result in very few extinction events in a scenario with no tracing and poor reporting of symptoms (see Figure 4 top-left), a larger outbreak size is required to distinguish between possible extinction events and exponential growth in scenarios with higher reporting and adherence and contact tracing interventions. As any scenario that does not go extinct will eventually reach 2000 cases, setting this threshold high only serves to prevent scenarios that do eventually go extinct from being missed.

Page 4: In the superspreading events, do you take into account the delays between developing symptoms and getting hospitalized?

We did take into account these delays, but in the end we felt that this section of the analysis was detracting from the main message of the manuscript and we have removed this section on superspreading events.

Section Additional observations: I don't really understand this idea of using the current outbreak size to say whether contact tracing can still control transmission. Do you mean here that the outbreak becomes too large for the public health system to perform the tracing? Theoretically, the tracing effectiveness does not depend on the initial number of individuals, but only on what proportion of onward infections can be prevented. Please clarify this.

Figure 4 shows that as the total number of cases so far (or the current outbreak size) increases, the probability of a large outbreak also increases. This is because of the overdispersion of the secondary case distribution. The higher the total number of cases, the higher the number of missed cases, and therefore the higher the number of missed super-spreaders for any given tracing effectiveness. At low case numbers, super-spreaders are rare and stochastic disease extinction is more likely.

Page 11, lines 380/381: I don't understand this sentence. Why should contacts of a negative testing person be isolated?

We agree that this phrasing was misleading, it was referring to a scenario where a contact was traced and isolated before the test result on the parent case was received. However, we have changed how we are modelling the testing and tracing process to better reflect the current UK strategy and this scenario is no longer relevant, hence it has been removed from the text.

Table 1 and Figure 5: The parameters you use for infection time are different than the ones reported by He et al (your ref 23) originally. Are these the values for the corrected analysis? However, they are also different from the parameters reported by Ashcroft in the Swiss Medical Weekly (doi:10.4414/smw.2020.20336). Could you please comment on this and justify your use of parameter values?

We acknowledge that we were not clear in the methods used to derive these values and have updated the Methods section to describe how these distributions were fitted:

“This Gamma distribution has been fitted under these sampling assumptions to serial interval data published by He et al. [24] using the fitdistr package in R and our resulting distributions qualitatively match those presented in the original paper (see Figure S1 in Supplementary File for more details).”

We have also included the code used to fit these distributions in the publicly available model code.

Figures 1 and 2: Please leave more space between upper and lower panels or add a horizontal line at zero for the upper panel.

Thank you for the suggestion, we have increased the spacing between panels and added additional axes and grid lines.

Figure 1a: From this figure I get the impression that you used a constant test sensitivity, not dependent on the time since infection. Is that correct? If not, I suggest to plot the test sensitivity you used in comparison to the Kurcika values. It is not clear to me from this figure, how this shift in testing was applied in relation to test sensitivity in the model. Please clarify.

We agree this figure was perhaps a clumsy representation of what we were trying to show around what time, relative to exposure, individuals were tested. This was mainly to justify the use of fixed sensitivity assumptions in the model. The figure is no longer included in the manuscript, but we have expanded upon the limitations of the fixed sensitivity assumption in the Discussion, as mentioned above.

Figure 1b: In the left panel, what does it mean for the line “no testing” that individuals leave quarantine after negative testing? Would individuals remain in quarantine when there is no testing? So is this exactly the same line as in the right panel for no testing? It looks to be somewhat lower. Can you clarify this in the figure caption?

We have shifted the focus of the manuscript to enhance relevance to the current situation and this figure no longer exists.

Figure 2: It might help the reader to include some gridlines or to add numbers to the medians of the boxplots. Now it is a bit hard to see how much higher or lower the plots are.

Good suggestion, we have added additional gridlines and axes to our plots to increase clarity.

Figure 3: I am not sure whether first hospitalization would be the trigger to start contact tracing or other measures. Why not detection of first symptomatic case? I would mention in the figure caption that a large outbreak is defined as 2000 cases.

We have shifted the focus of the manuscript to enhance relevance to the current situation and this figure no longer exists.

Figure 4: Nice overview over contact tracing process!

Thank you very much!

REVIEWERS' COMMENTS

Reviewer #1 (Remarks to the Author):

I sincerely appreciate the careful thought the authors put into the reviewer feedback. My major concerns have all been addressed. I think the new title, abstract and overall messaging is much improved.

A general suggestion is that there are several sentences that combine multiple thoughts together – sometimes with multiple sets of 'and'. Some proofreading for clarity will likely be helpful.

Some more specific suggestions:

- cWhen first used in text (and possibly in abstract as well), define what is meant by 'imperfect adherence' and 'imperfect reporting'.
- Figure 2 legend – this figure has no 'dashed' line
- Figure 2 – How can contact tracing leading to negative values for the % reduction of R? Perhaps explain in legend
- Figure 3 – The two-day delay appears to be red, but legend labels it as green. Also, should 'and $R_s = 1.3$ ' be deleted since two values of R are considered?

Reviewer #2 (Remarks to the Author):

The authors have responded to all my comments and I am satisfied with their responses. I still believe that including a test sensitivity that varies with time since exposure would have strengthened the analysis, but I see that this would be much additional work.

I saw one typo: in the caption of figure 3 the lines for 2-day delay are red (it says green).

Reviewer #1 (Remarks to the Author):

I sincerely appreciate the careful thought the authors put into the reviewer feedback. My major concerns have all been addressed. I think the new title, abstract and overall messaging is much improved.

A general suggestion is that there are several sentences that combine multiple thoughts together – sometimes with multiple sets of ‘and’. Some proofreading for clarity will likely be helpful.

Response: We had modified some of the longer sentences to improve clarity.

Some more specific suggestions

- When first used in text (and possibly in abstract as well), define what is meant by ‘imperfect adherence’ and ‘imperfect reporting’.

Response: We have removed these terms as they were not frequently use in the manuscript.

- Figure 2 legend – this figure has no ‘dashed’ line

Response: Good spot – legend has been adjusted to correct description.

- Figure 2 – How can contact tracing leading to negative values for the % reduction of R? Perhaps explain in legend

Response: We have made this clear in the legend that negative values represent stochastic fluctuation.

- Figure 3 – The two-day delay appears to be red, but legend labels it as green. Also, should ‘and $R_s = 1.3$ ’ be deleted since two values of R are considered?

Response: Thank you for pointing this out, you’re correct and we have adjusted the legend accordingly.

Reviewer #2 (Remarks to the Author):

The authors have responded to all my comments and I am satisfied with their responses.

I still believe that including a test sensitivity that varies with time since exposure would have strengthened the analysis, but I see that this would be much additional work.

I saw one typo: in the caption of figure 3 the lines for 2-day delay are red (it says green).

Response: We have corrected the caption to match the figure colours.